# DrNote: An open medical annotation service

**Johann Frei**[1]*, **Iñaki Soto-Rey**[2‡], **Frank Kramer**[1]

**1** IT-Infrastructure for Translational Medical Research, Faculty of Applied Computer Science, University of Augsburg, Augsburg, Germany, **2** Medical Data Integration Center, Institute for Digital Medicine, University Hospital Augsburg, Augsburg, Germany

‡ Clinical Partner
* johann.frei@informatik.uni-augsburg.de

## Abstract

In the context of clinical trials and medical research medical text mining can provide broader insights for various research scenarios by tapping additional text data sources and extracting relevant information that is often exclusively present in unstructured fashion. Although various works for data like electronic health reports are available for English texts, only limited work on tools for non-English text resources has been published that offers immediate practicality in terms of flexibility and initial setup. We introduce DrNote, an open source text annotation service for medical text processing. Our work provides an entire annotation pipeline with its focus on a fast yet effective and easy to use software implementation. Further, the software allows its users to define a custom annotation scope by filtering only for relevant entities that should be included in its knowledge base. The approach is based on Open-Tapioca and combines the publicly available datasets from WikiData and Wikipedia, and thus, performs entity linking tasks. In contrast to other related work our service can easily be built upon any language-specific Wikipedia dataset in order to be trained on a specific target language. We provide a public demo instance of our DrNote annotation service at https://drnote.misit-augsburg.de/.

## Author summary

Since much highly relevant information in healthcare and clinical research is exclusively stored as unstructured text, retrieving and processing such data poses a major challenge. Novel data-driven text processing methods require large amounts of annotated data in order to exceed non data-driven methods' performance. In the medical domain, such data is not publicly available and restricted access is limited due to federal privacy regulations. We circumvent this issue by developing an annotation pipeline that works on sparse data and retrieves the training data from publicly available data sources. The fully automated pipeline can be easily adapted by third parties for custom use cases or directly applied within minutes for medical use cases. It significantly lowers the barrier for fast analysis of unstructured clinical text data in certain scenarios.

**Data Availability Statement:** The Wikipedia and WikiData datasets are publicly available at: https://dumps.wikimedia.org/enwiki/ https://dumps.wikimedia.org/dewiki/ https://dumps.wikimedia.org/wikidatawiki/entities/ Our project repository is

publicly available at: GitHub: https://github.com/frankkramer-lab/DrNote.

**Funding:** This work is a part of the DIFUTURE project funded by the German Ministry of Education and Research (Bundesministerium für Bildung und Forschung, BMBF) grant FKZ01ZZ1804E. The funders had no role in study design, data collection and analysis, decision to publish, or preparation of the manuscript.

**Competing interests:** The authors have declared that no competing interests exist.

## Introduction

Effective processing of natural clinical language data has increasingly become a key element for clinical and medical data analysis. Recent trends in the field of natural language processing (NLP) have established novel data-driven neural approaches to largely improve a broad variety of language and text analysis tasks like neural machine translation, text summarization, question answering, text classification and information extraction in general. Most notably, emerging from semantic word embeddings like Word2Vec [1] and GloVe [2], contextualized word embedding techniques like ELMo [3] or BERT [4] based on the Transformer network architecture [5] are applied in order to solve most of context-specific downstream tasks. Attention-based language models therefore gained popularity among the NLP research community since they are able to outperform simpler rule-based models, statistical methods like conditional random fields and other, neural methods like LSTM-based models on core NLP tasks such as named entity recognition (NER).

On the matter of domain-specific neural approaches for NLP numerous derivatives [6–11] are applied for various NLP downstream tasks. The trend of these neural approaches appear to steer towards end-to-end models [12] which are often optimized for specific purposes [13]. While most works focus on English data, creating cross-lingual approaches [4, 14, 15] for medical applications is difficult due to the lack of sufficient data.

Traditional non-deep learning NLP systems often adopt pipeline-based approaches [16, 17] for text processing in which each pipeline stage performs a modular text processing task, enabling the reuse of single components on different applications and contexts in a simplified fashion. The core components often rely on feature-based machine learning or linguistic rule-based methods, although certain frameworks [16, 18] integrate also neural approaches for certain NLP tasks in more recent versions. For the framework of [18], a domain-specific model [19] has been published for biomedical applications for English text data. For German texts, mEx [20] implements a similar pipeline for clinical texts based on SpaCy [16], albeit its trained models have not been published.

Historically, NLP software for medical applications has been an ongoing research subject. The software system *medSynDiKATe* [21] is an early approach to extract relevant information from pathology finding reports in German language. *Apache cTAKES* [22] is another modular software for medical text processing, following the UIMA architecture, that uses OpenNLP [23] for text analysis. While [22] is mainly designed for English texts, [24] shows only moderate results for German data when using input text translation into English. *HITEx* [25] based on the *GATE* [26] framework, and *MetaMaps* [27] present comparable notable implementations for medical text processing for English text data. Provided as a public web API, *PubTator* [28] is a similar text mining tool for English biomedical text annotations with support for a fixed set of entry types.

From the perspective of commercial software for medical text analysis in German language, *Averbis Health Discovery* [29] provides an industry solution to NLP tasks for clinical applications. For a deeper insight in remaining challenges of non-English medical text processing we point to the review paper [30]. More information on the situation of clinical text analysis methods such as for medical concept extraction and normalization or for clinical challenges in general are presented in review papers [31–33]. In similar contexts, *Trove* [33] is proposed as a framework for weak supervised clinical NER tasks. While the latter work yields a broad overview on key aspects of different methodological concepts and covers weak supervised settings with ontology-based knowledge bases in English, it acknowledges the need for further work on non-English contexts.

For text annotation and entity linking in general, earlier works focus on Wikipedia and WikiData as knowledge base. Entity linking on unstructured texts to Wikipedia was shown in [34–37], even before the WikiData [38] knowledge graph was introduced. Different entity linking approaches were evaluated and compared in [39]. In addition to WikiData, other knowledge bases [40–43] have been released as well. For tagging engines like *TagMe* [37], refined entity linking systems [44, 45] were released. More recently, neural-based entity linking methods have been proposed [46–48].

## Motivation

By considering common natural language processing tasks as a learning problem, this inherently implies the need for training data. Since novel Transformer-based architectures have been proven effective on large amount of domain-specific training data [6–11], training such domain-specific models for certain languages from scratch without any pretraining [7, 9, 10] remains a major challenge due to the lack of appropriate datasets in general. Hence, transfer learning approaches are commonly used for use case-specific downstream tasks and integrated in practical application [13], in order to mitigate the required amount of training data and boost the performance of the model.

Open datasets of biomedical texts and clinical letters for English languages have been published [49, 50]. In the particular case of German data resources for clinical letters, the situation is more dire [30, 51, 52] as no large dataset is publicly available.

In addition, one property of natural language processing methods concerns the possible dependency on one specific language: Although works on cross- and multilingual language models like XLM, XLM-R [14, 15] or mBERT [4] present notable results, they indicate higher downstream task performance scores for monolingual models on non low-resource languages.

Since text processing pipelines need to be manually fine tuned for their corresponding downstream task on aggregated training data in order to reach significant level of performance, these pipelines require a high level of technical skill sets in order to apply existing methods based on contextualized word embeddings in dedicated domain contexts.

## Contributions

Consequently, in this work we primarily focus on methods that do not rely on techniques like contextualized word embeddings and can be built using a public dataset. From our perspective, this enables a simplified process for build, deployment and application.

This work presents an open annotation tool for unstructured medical texts which implements an entity linking solution.

Our key contributions can be considered as an ensemble of the following items:

- *Automated build process:* Our annotation tool requires precomputed annotation data in order to perform the entity linking tasks. To preprocess and obtain the annotation data, we provide a fully automated, end-to-end build pipeline that enables the user to adapt our pipeline for custom use cases.

- *Use of public data:* The annotation tool relies on the publicly available, open WikiData and Wikipedia datasets. The datasets are used for initial training of the annotation candidate classifier at build time, and as a knowledge base for entity linking later during the text annotation.

- *Language support:* Our implementation is capable of adopting other languages in its build pipeline. The user can choose a specific language from the set of supported languages in Wikipedia.

- *Usability:* The annotation service offers a simplified RESTful API for entity linking. The user can input plain text into a basic web interface. In addition, PDF documents can be uploaded and processed. In the case of using pretrained annotation data, the annotation service can be easily deployed on premise instantly.

## Materials and methods

### Open datasets

The need for training data is one of the major issues in the area of data-driven methods. In this work, we combine two open public data sets in order to retrieve appropriate training data.

**WikiData.** WikiData [38] is a free open knowledge base with multilingual, structured data. Its entities (*items*) are represented in a graph structure, in which each entity consists of an item identifier and main item label. These items can store short *description* texts, *labels* and potential *alias* labels for certain languages. In addition, an item may comprise a list of *statements* to further encode knowledge. Hereby, a statement is defined by a property and a list of corresponding values. These values can either encode explicit structured values or references to other entities in the knowledge base. Furthermore, each item can store references to other wiki entries through its *sitelink* attribute. Given by the nature of its graphical representation, the WikiData repository can be queried through a public SPARQL-API. The entire WikiData knowledge base is also accessible through a file download for local use.

**Wikipedia.** For the sake of simplicity, the Wikipedia platform is considered in this work as a set of independent open public language-specific wiki sites. Each wiki site is composed by a set of wiki pages. A wiki page consists of a page title and the page content. The page content is written in the *Wikitext* syntax which constitutes a simplified hypertext markup language. Plain texts from a wiki page contain words that can reference other wiki pages, and thereby form a graph-like structure consisting of one node per wiki page. Every wiki page references a corresponding item from the WikiData knowledge base. These references often expose the limitation of linking wiki pages to WikiData items due to the diverging concept scope granularity: For instance, the German wiki page for *Diabetes mellitus* links to the WikiData item *Q12206* which in reverse links back to the German *Diabetes mellitus* wiki page through the entity sitelinks. However, the WikiData item *Q3025883* represents the concept of *Type-2-Diabetes* and its sitelink back to the German wiki page resolves to the wiki page *Diabetes mellitus* with focus on the page section *Diabetes Typ 2*. This implies a potential loss of information due to the granularity mismatch since the mapping between wiki pages and WikiData items does not exhibit the bijective property.

In general, the language-specific wiki dataset can be downloaded in order to obtain the content of all wiki pages.

### Text annotation

One of the decisive components that are vital to such an annotation pipeline is the tagging and linking component. The high-level task for this component is to identify all semantically corresponding entities of a given knowledge base for a given text and link the affected text positions by their related entity references. This task is regarded as an entity linking (EL) task.

$$\omega_s = \delta[s], \epsilon_s = link(\omega_s), \epsilon_s \in \Omega, \tag{1}$$

where:

$\omega$ = mention

$\delta$ = text document

$s$ = identified text span

$\epsilon$ = (concept) entity

$\Omega$ = knowledge base (KB)

$link(m)$ = entity linking function

The objective to approximate the entity linking function $link(m)$ can be decomposed into two independent subtasks. The first step covers the mention detection and candidate generation. At this step, all possible entities $\epsilon$ from the knowledge base $\Omega$ that match to the detected mention $\omega$ are considered. The second step regards the selection of the best entity candidate for the mention $\omega$ (Entity Disambiguation). The proper design of this step heavily depends on the $\omega \leftrightarrow \epsilon$ match scoring function which may incorporate context-dependent scores in addition to context-independent similarity metrics.

In this work, we heavily rely on OpenTapioca [53] for solving the entity linking objective. OpenTapioca leverages the tagging functionality of the Apache Solr software in order to implement the candidate generation step. OpenTapioca creates and prepares a Solr collection in advance to index all relevant terms of the WikiData knowledge base for accelerated mention lookup and tagging.

For the estimation of the matching score of a mention to a corresponding entity candidate, the following local feature vector is defined and sampled for each pair of entity candidate $\epsilon$ and mention $\omega$:

$$F(\omega, \epsilon) = (-\log\ p(\omega), \log\ PR(\epsilon), stmd(\epsilon), sl(\epsilon), 1) \tag{2}$$

where:

$p(\omega)$ = probability of mention $\omega$ in the language model

$PR(\epsilon)$ = PageRank score of entity $\epsilon$ in KB $\Omega$

$stmd(\epsilon)$ = number of statements of entity $\epsilon$

$sl(\epsilon)$ = number of sitelinks of entity $\epsilon$

Hereby, a feature matrix $FM_0$ is constructed from the stacked local feature vectors. The sequence of detected mentions forms a weighted graph $G$, where each ($\omega$-$\epsilon$) pair yields a node, connected to all nodes with neighboring $\omega_{neighbor}$ mentions. To represent semantic clusters, the connection weights between two nodes a, b with their entities $\epsilon_b$ and $\epsilon_a$ are modulated by the probability to reach each other or a common entity $\epsilon_{3rd}$ in the knowledge base graph $\Omega$. The reachability is limited to first order connectivity. The feature matrix $F_0$ is propagated along the stochastic adjacency matrix $\tilde{M}$ for $n$ iterations, where $M$ represents the unnormalized (non-stochastic) adjacency matrix of the graph $G$, resulting in the contextualized feature tensor ($FM_0, FM_1, \cdots, FM_n$). A support vector machine (SVM) is applied to estimate the score for each entity candidate of each mention. For a deeper explanation, we point to the original paper [53].

Whereas the uni-gram language model is computed on the WikiData entity labels as an approximation, and the Page Rank scores for the entities can be computed on the WikiData knowledge base graph structure, it is important to note that the SVM classifier cannot be trained without annotated training data. Thus, the user is required to extract documents and annotate the documents' mentions with their related entity links manually.

OpenTapioca requires a tool-specific *profile* for term indexing that defines all entities which should be part of the knowledge base $\Omega$. The text annotation only covers terms that were previously included in the OpenTapioca *profile*.

### Text postprocessing and filtering

In order to allow encoding of prior knowledge about the target annotation structure, it is necessary to analyse the input text data by an NLP toolchain. We rely on the library SpaCy [16] in conjunction with its published pretrained pipeline components. The components mainly consist of a simple universal part-of-speech tagger, a more detailed part-of-speech tagger, a morpholgizer and a dependency parser.

## Results

### Build pipeline

An automated pipeline has been developed that performs the following steps:

**Installation of required components.** All required dependencies are installed to the local machine. This mainly includes a container runtime and basic shell tools.

**Automated NIF extraction.** We acquire the required training data through a pipeline of parsing, extraction and transformation steps of the formerly mentioned datasets. First, the Wikipedia pages for a given language code are parsed. During that step, all referencing terms in the page texts are extracted and their referenced pages are resolved. Given the OpenTapioca *profile*, we then query the SPARQL API to obtain all affected WikiData items. The entire WikiData dataset is parsed and for each WikiData item sitelink that points to its Wikipedia page, we add its item identifier to the processed Wikipedia page in the database to establish a bijective mapping while ignoring more fine-granular WikiData items.

Given the parsed and linked data for WikiData and Wikipedia, we can select all pages that contain referencing terms in the text to other pages if the related WikiData items were included by the OpenTapioca *profile*. By doing so, we treat relevant referencing terms as word annotations and, therefore, can synthesize the required dataset with annotated mentions. To avoid irrelevant text sections, only the sentences with relevant terms are further extracted. For sentence splitting, SpaCy [16] is used. The transformed data is stored in the common NIF format.

**OpenTapioca annotation setup.** The initialization steps for OpenTapioca are performed as follows: Based on the given OpenTapioca *profile*, all labels and alias terms of the selected WikiData items are loaded and indexed by the Solr instance. In addition, the buildup of the entity graph for the PageRank computation as well as the buildup of the uni-gram language model is run, followed by the training of the SVM classifier on the extracted NIF dataset. The logical data flow process is visualized in the box *Build Stage* of Fig 1.

The build pipeline eventually outputs a single package file that is needed for the instant deployment of the annotation service.

### Service

The annotation service provides a platform for text input processing through an HTTP-based RESTful API as well as through a basic web interface. The implementation integrates the annotation strategy of OpenTapioca and the entity tagging process through an Apache Solr instance. For a successful deployment, the prebuilt file package is required which stores the classifier and language model information as well as the index database of the Solr collection for accelerated entity lookup.

In addition to the plain text annotation, the service features the processing of PDF documents with options for data input and output. Concerning the data input, the text from a PDF document is extracted in order to apply the entity linking task. Therefore, PDF documents with digital text information can be directly processed. In case of scanned documents that

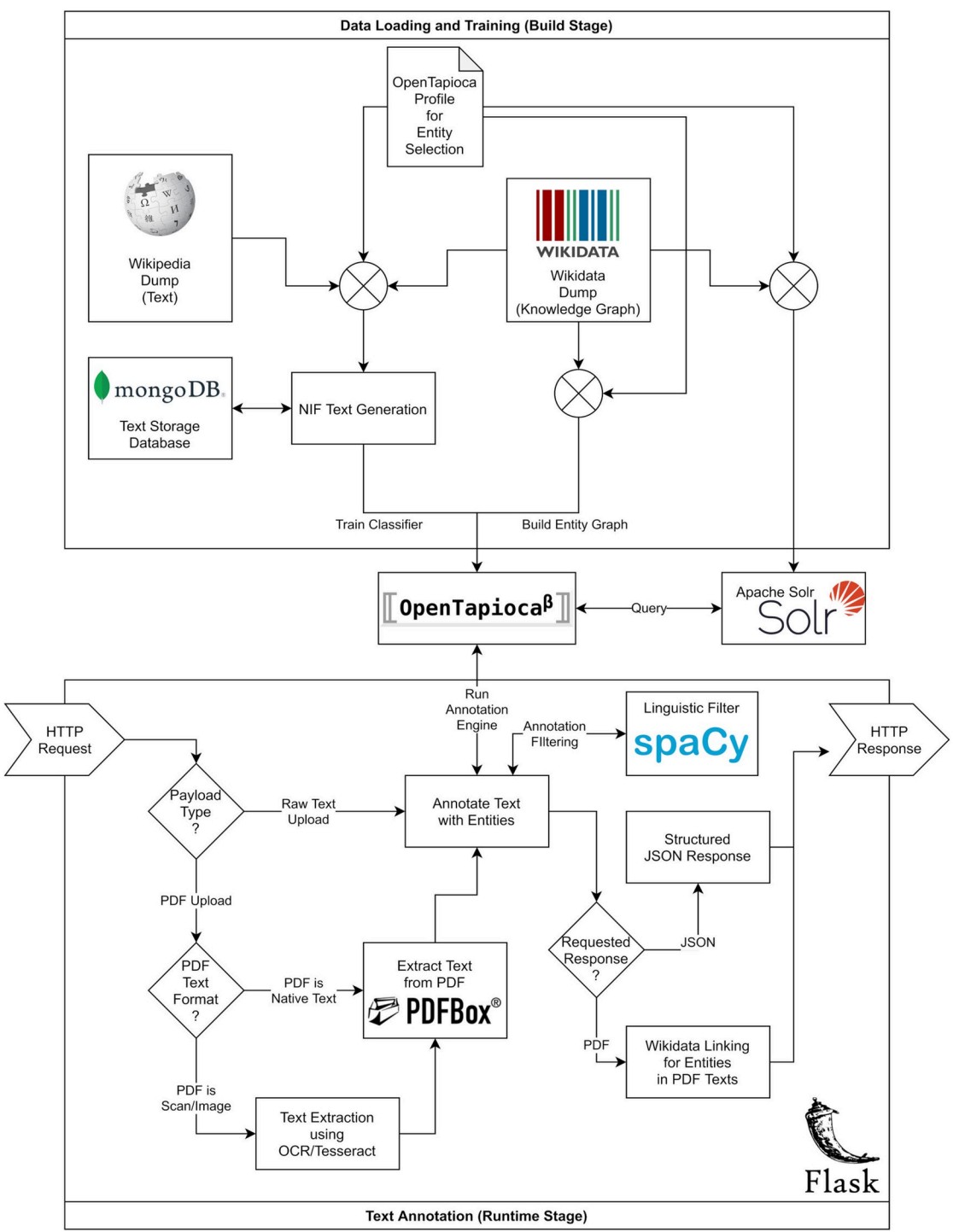

**Fig 1. Visualization of logical data flow: The box "Build Stage" describes the components for data generation and preparation.** The box "Runtime Stage" illustrates the processing of an annotation request. The components of OpenTapioca and Apache Solr are shared during build and runtime stage.

encode their information in an image format, an additional OCR step is applied. The OCR step is based on the Tesseract [54] software.

Concerning the data output, the annotation information can be either provided as a machine-readable JSON response or as a PDF document with embedded hyperlinks to the corresponding WikiData item page for all identified mentions. The service provides the option to postprocess the input text and its annotations in order to filter out annotations which are implausible based on their linguistic or structural properties. For instance, one may require the annotations to have at least one word token tagged as a noun in certain scenarios.

The logical data flow for the annotation service is depicted in the box *Runtime Stage* of Fig 1.

### Build for medical use case

We apply the developed build pipeline and annotation service for our central use case for medical text analysis. The computed initialization data for the annotation service was retrieved by our build pipeline for our specified use case.

One of the most relevant information in medical letters includes data which is associated to symptoms, diagnoses, drugs and medications. Therefore, the entity selection process is managed in the way to cover all WikiData items that represent direct or indirect instances of these concepts in the knowledge base.

Our strategy to select all relevant entries leverages the graphical structure of the WikiData knowledge base. An item is a part of the knowledge base index if at least one of the following conditions is satisfied:

- The item has a *Disease Ontology* (P699) statement entry.

- The item has an *UMLS CUI* (P2892) statement entry.

- The item has a *MeSH descriptor ID* (P486) statement entry.

- The item has a *MeSH tree code* (P672) statement entry.

- The item is a subclass of *Medication* (P12140).

Since the entity linking task can only detect references to entities that have been indexed through the build pipeline, an effective feature selection contributes crucially to the capabilities of our annotation service. In our medical use case scenario, the need for multiple selection features can be demonstrated by the fact that the WikiData knowledge base can be considered incomplete. For instance, 27786 unique entities with an associated UMLS CUI statement can be found in WikiData at the time of writing. In contrast, the UMLS metathesaurus (2020AB) consists of 15938386 total entries and 4413090 unique CUIs. Adding the *MeSH descriptor ID* (P486) to the UMLS CUI selection feature increases the number of entries by 26932 and therefore can add highly relevant items to the knowledge base despite the problem of missing WikiData UMLS references.

Using multiple direct features of an item for entity selection, however, only mitigates the described issue. In addition to such direct features, we demonstrate that utilizing the hierarchical internal WikiData structure can be beneficial in order to further reduce the item miss rate due to the lack of data or incomplete data in WikiData items: The item *Medication* (Q12140) is referenced by several other items through the property *Subclass of* (P279), and thus, all items that are a subclass of *Medication* can be directly selected as relevant entries. In this context, not only first order subclass items are included but also all n-degree subclass items from the WikiData hierarchy. For instance the item *Opioid* (Q427523) is selected through the hierarchy path *Medication* $\subseteq$ *Analgesic* $\subseteq$ *Opioid* where $\subseteq$ is a *Subclass of* reference.

**Table 1. Build Processing Times.**

| Stage | Substage | Time | Multicore | Profile |
|---|---|---|---|---|
| NIF & OpenTapioca | Data download | 5h | no | independent |
| NIF | Page & redirect extraction | 2h | no | independent |
| NIF | Entity extraction | 16h | yes | independent |
| NIF | Entity filtering | 24m | yes | independent |
| NIF | Pagelinks extraction | 106h | yes | independent |
| NIF | NIF file generation | 1h | no | dependent |
| OpenTapioca | Language model creation | 16h | no | independent |
| OpenTapioca | Link extraction | 15h | no | independent |
| OpenTapioca | Link sorting | 45s | no | independent |
| OpenTapioca | Link sparse matrix conversion | 24m | no | independent |
| OpenTapioca | Page rank computation | 26m | no | independent |
| OpenTapioca | Entity indexing | 11h | no | dependent |
| OpenTapioca | Classifier training | 2m | partly | dependent |

Build times for German medical use case: NIF-based substages interact with a MongoDB database. MongoDB supports read operations on multiple cores. Profile-dependent stages require recomputation on profile changes.

The build pipeline was executed for an German OpenTapioca profile with the formerly mentioned item selection features. The processing and training was performed on an 8-core Intel Xeon Silver 4210 virtual machine with 128GB memory. The computation times for various pipeline substages are depicted in Table 1. While later substages depend on the defined OpenTapioca profile and require recomputation on profile changes, earlier substages are independent of profile changes. Regarding multicore scaling, parts of the pipeline support the multiprocess architecture model. The computation times vary based on the number of WikiData entities and Wikipedia pages. For the presented computation times, the NIF generation stage processed 95.1M WikiData entities and 5.6M German Wikipedia pages.

As described in the previous chapter, our annotation service exposes a simple HTTP REST interface as well as a graphical web interface. The web interface is depicted in Fig 2 for an in-browser text annotation of an anonymized text snippet from the MIMIC-III [49] dataset. For demonstration purposes, we created a PDF document with the same text content and submitted the document to the PDF upload interface to retrieve an annotated PDF document with its embedded annotation links as output. The result is shown in Fig 3.

Our pretrained data for an instant service deployment as well as the source code is available at our project repository page at https://github.com/frankkramer-lab/DrNote.

## Performance evaluation

To evaluate the annotation performance we compare our method with Apache cTAKES (version 4.0.0.1) and PubTator (https://www.ncbi.nlm.nih.gov/research/pubtator/api.html) as baseline. Since our method is designed for multilingual use cases and for non-English data in specific, we focus on German text data for performance comparisons. To avoid inadequate evaluation issues such as missing UMLS references or ambiguous mappings between non-isomorphic knowledge bases, we consider the annotation task as a binary text segmentation task at which the annotation spans define the binary segmentation mask. For clinical contexts, we randomly drew 50 samples from the GERNERMED [55] test set and manually corrected incorrect annotation spans, since the dataset is based on an automated translation of the *n2c2 2018*

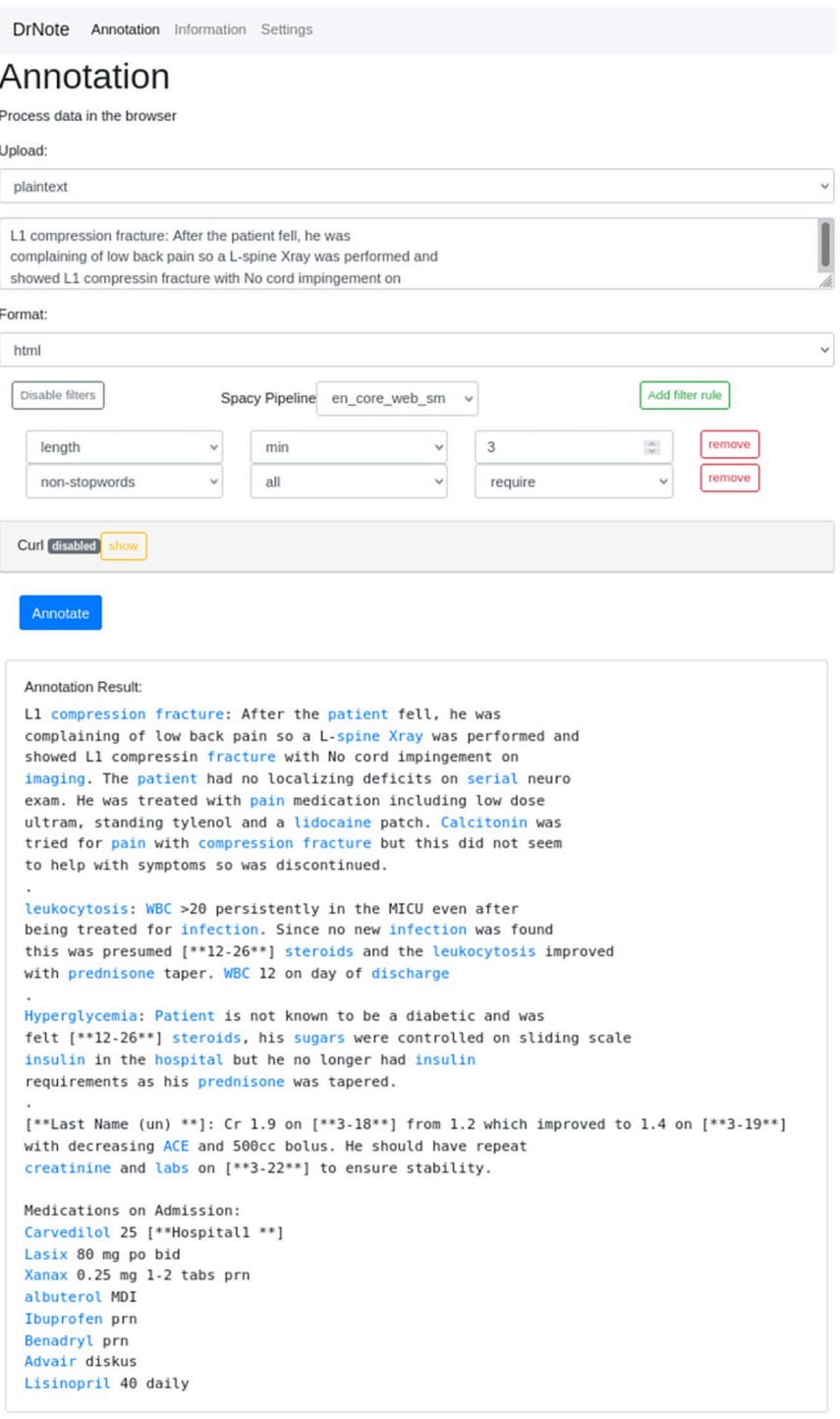

**Fig 2. Demo for web interface.** Browser-based annotation on example data from the MIMIC-III [49] dataset.

*ADE and Medication Extraction Challenge* [50] dataset with automated annotation alignments. All labels except for *Drug* were omitted for comparison reasons.

In order to quantify the domain-shift bias in non-clinical contexts on the biomedical *Mantra GSC* [56] datasets. In these datasets, the annotations are linked to their corresponding

L1 compression fracture: After the patient fell, he was
complaining of low back pain so a L-spine Xray was performed and
showed L1 compressin fracture with No cord impingement on
imaging. The patient had no localizing deficits on serial neuro
exam. He was treated with pain medication including low dose
ultram, standing tylenol and a lidocaine patch. Calcitonin was
tried for pain with compression fracture but this did not seem
to help with symptoms so was discontinued.

.
leukocytosis: WBC >20 persistently in the MICU even after
being treated for infection. Since no new infection was found
this was presumed [**12-26**] steroids and the leukocytosis improved
with prednisone taper. WBC 12 on day of discharge

.
Hyperglycemia: Patient is not known to be a diabetic and was
felt [**12-26**] steroids, his sugars were controlled on sliding scale
insulin in the hospital but he no longer had insulin
requirements as his prednisone was tapered.

.
 [**Last Name (un) **]: Cr 1.9 on [**3-18**] from 1.2 which improved to 1.4 on
[**3-19**]
with decreasing ACE and 500cc bolus. He should have repeat
creatinine and labs on [**3-22**] to ensure stability.

Medications on Admission:
Carvedilol 25 [**Hospital1 **]
Lasix 80 mg po bid
Xanax 0.25 mg 1-2 tabs prn
albuterol MDI
Ibuprofen prn
Benadryl prn
Advair diskus
Lisinopril 40 daily

**Fig 3. Demo for PDF annotation.** A PDF page demo with embedded annotations on example data from MIMIC-III [49] dataset.

UMLS entries. Since WikiData lacks large parts of the UMLS references mentioned (Medline: 90 out of 309 UMLS concepts known, EMEA: 121 out of 425 UMLS concepts known), the DrNote scores are also evaluated on a filtered set of UMLS annotations that are reference in WikiData, yet in all setups, DrNote annotations were limited to entities that are subclasses

**Table 2. Annotation Performance Evaluation.**

| Dataset | Method | F1 score |
|---|---|---|
| GERNERMED | cTAKES | 0.632 |
| GERNERMED | DrNote | **0.722** |
| GERENRMED | PubTator | 0.523 |
| Medline GSC | cTAKES | 0.148 |
| Medline GSC | DrNote | **0.226** |
| Medline GSC | PubTator | 0.123 |
| EMEA GSC | cTAKES | 0.162 |
| EMEA GSC | DrNote | **0.261** |
| EMEA GSC | PubTator | 0.0728 |
| Medline GSC | DrNote (filtered) | 0.414 |
| EMEA GSC | DrNote (filtered) | 0.503 |

Evaluation results of cTAKES, PubTator and DrNote (ours) on various datasets. Filtered results exclude annotations from the ground truth if their corresponding UMLS CUI is not referenced in WikiData.

(P279) or instances of (P31) of medications (Q12140) for comparison reasons. Apache cTAKES uses the UMLS metathesaurus directly and therefore does not suffer from incomplete UMLS data. Given its focus on biomedical texts, PubTator supports the entity concepts *gene*, *disease*, *chemical*, *species*, *mutation* and *cellline*. The entity concept chemicals is identified by PubTator through a search-based dictionary lookup in the MeSH thesaurus.

The evaluation results are given as f1 scores based on the character-level text segmentation masks from the ground truth (GT) and the predicted segmentation in Table 2.

While our method exhibits substantially better text segmentation f1 score performance in comparison to cTAKES and PubTator, and demonstrates considerable results on the clinical dataset, all methods show subpar results on biomedical datasets. We mainly attribute this circumstance to the fact that both biomedical datasets include annotation phrases in part-of-speech (PoS) forms other than nouns, in which case both methods tend to fail. While cTAKES and PubTator are incapable of German word stemming due to the focus on English, our method relies on the nominalized WikiData labels and fails for similar reasons. However, our method seems to perform better on all datasets which we attribute to the broader set of common alias labels in WikiData compared to the related UMLS or MeSH entry labels as illustrated in Fig 4. In contrast to cTAKES, our method can also use linguistic information to avoid obvious PoS-related annotation errors as shown in Fig 5. cTAKES is still able to detect certain German UMLS entities due to the fact that the German language represents the largest non-English language in the UMLS metathesaurus, however PubTator uses the English MeSH database and does not include the German MeSH terms. Conversely, PubTator is able to detect specialized codes from MeSH whereas cTAKES does not detect certain codes (Fig 6) although the displayed code *RAD001* is present in the UMLS database. With respect to our method, we also identified scenarios in which correct annotations were skipped due to our filter mechanism meant to exclude non-medication items as shown in Fig 7. For this particular instance, the item *Steroid* had been annotated correctly, yet due to the structure in the WikiData graph, the item (Q177911) is not classified as being a subclass or instance of medication and skipped for that reason.

## Discussion

Considering the dire state of natural language processing tools with support for multi-language data input in the medical context, the presented annotation service can offer useful services for

Bei ihrer Ankunft in der MICU wurde sie mit Laktulose und Rifaximin behandelt.

- DrNote
- CTAKES
- GT
- DrNote
- Pubtator
- GT

Sie wurde mit einem IV-Protonenpumpenhemmer gestartet und ihr

- DrNote
- GT

**Fig 4. Comparative Sample 1 and 2.** Specialized German term missing in UMLS or MeSH, (*Laktulose*,*Protonenpumpenhemmer*, cTAKES & PubTator).

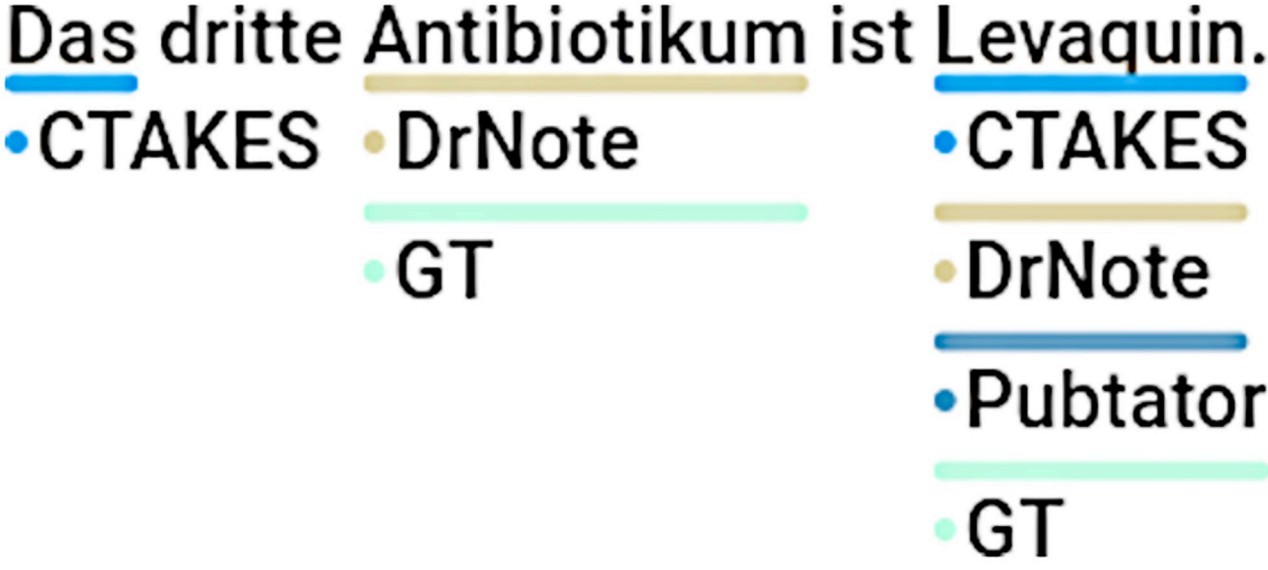

**Fig 5. Comparative Sample 3.** Artifacts from lack of German linguistics (*Das*, cTAKES).

- 1988-03-02 RAD001 5 mg po - 2010-03-07 Start PKC412 50 mg po bid.

- DrNote
- Pubtator
- GT

- DrNote
- Pubtator
- GT

**Fig 6. Comparative Sample 4.** Weakness of cTAKES to detect certain codes (*RAD001*, cTAKES).

## Sie wurde von ihrem Steroid-Medikament entwöhnt.
- CTAKES
- Pubtator
- GT

**Fig 7. Comparative Sample 5.** Ignored annotations due to WikiData graph structure (*Steroid* as no subclass/instance of medication), DrNote).

research applications and related text analysis tasks. Inherently to the chosen dictionary-based entity detection and linking approach, the capabilities of the service as well as its limitations exclude the tool for certain tasks: The entity detection method is limited to only recognize entities that are part of the WikiData knowledge base label and alias term sets, excluding semantically related terms or slightly altered, corrupted terms that are closely related to their correct term when evaluated on the Levenshtein distance metric. A potential remedy for this issue can be the use of a spellchecker component. The entity disambiguation step cannot reject false positive mentions in situations where the entity candidate presents imprecise label or alias values. For instance, the item *Universe* (Q1) contains the word *all* as an alias value. Subsequently, *all* will be linked to Q1 in the case that Q1 was previously included by the entity selection step. To effectively counter such artifacts, a deeper semantic understanding is required. However, the buildup of semantic understanding is mostly handled by utilizing large training data from certain target domains and poses a major disadvantage of data-driven methods since large datasets can be challenging to obtain and may jeopardize robust multi-language support. By offering advanced annotation filter rules based on linguistic features through SpaCy, this may alleviate the problem in situations where a pretrained SpaCy pipeline is available for the corresponding text language.

Currently, our approach is strictly tied to the label and alias terms from WikiData that are typically nominalized. Therefore, relevant terms in different part-of-speech configurations such as adjectives cannot be detected due to the lack of language-dependent stemming or lemmatization. Further work on such improvements is considered future work.

The synthesis of the annotated dataset of relevant mentions from the Wikipedia and WikiData datasets and its transformation into the NIF file format only considers links of mentions at a page to another referenced page whenever the link was inserted manually by a Wikipedia author. In frequent cases, only the first mention of a referenced concept is linked by the authors on the page, despite the fact that the mention text may appear multiple times on the same Wikipedia page. This may induce lower recall scores in contrast to a complete and manual annotation. Our mitigation approach reduces the probability of including false negative terms by only extracting single sentences from the Wikipedia page texts.

## Conclusion

In this work we introduced our annotation service DrNote as an open platform for entity linking in the context of medical text processing with multi-language support. The annotation service can operate directly on precomputed initialization data which are provided for instant deployment. An fully automated build pipeline was presented to enable users to customize the annotation service for specific needs while the generated dataset solely relies on open public

data. We presented the feature support for PDF document processing and annotation as well as the integration of SpaCy for advanced linguistic-based annotation filtering.

Common limitations of the chosen entity linking approach were further discussed as well as its conceptual drawback compared to competing data-driven approaches. While purely data-driven approaches may enable huge advancements over traditional approaches, their individual applicability for certain languages in the medical context remains to be challenging due to the lack of sufficiently large training data. Privacy concerns and legal restrictions for data use and access may hinder further improvements on the availability of such datasets in the future.

## Author Contributions

**Conceptualization:** Johann Frei.

**Data curation:** Johann Frei.

**Funding acquisition:** Frank Kramer.

**Investigation:** Johann Frei.

**Methodology:** Johann Frei.

**Project administration:** Frank Kramer.

**Resources:** Frank Kramer.

**Software:** Johann Frei.

**Supervision:** Iñaki Soto-Rey.

**Validation:** Johann Frei.

**Writing – original draft:** Johann Frei.

**Writing – review & editing:** Iñaki Soto-Rey, Frank Kramer.

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
