## [Decision Letter · Decision Letter 0]

13 Sep 2021

PDIG-D-21-00039DrNote: An open medical annotation servicePLOS Digital Health

Dear Dr. Frei,

Thank you for submitting your manuscript to PLOS Digital Health. After careful consideration, we feel that it has merit but does not fully meet PLOS Digital Health’s publication criteria as it currently stands. Therefore, we invite you to submit a revised version of the manuscript that addresses the points raised during the review process.

EDITOR: Please insert comments here and delete this placeholder text when finished. Be sure to:Indicate which changes you require for acceptance versus which changes you recommendAddress any conflicts between the reviews so that it's clear which advice the authors should followProvide specific feedback from your evaluation of the manuscript

Please ensure that your decision is justified on PLOS Digital Health’s publication criteria and not, for example, on novelty or perceived impact.

We look forward to receiving your revised manuscript.

Kind regards,

Imon Banerjee

Section Editor

PLOS Digital Health

Journal Requirements:

1. We ask that a manuscript source file is provided at Revision. Please upload your manuscript file as a .doc, .docx, .rtf or .tex. If you are providing a .tex file, please upload it under the item type ‘LaTeX Source File’ and leave your .pdf version as the item type ‘Manuscript’.

2. We do not publish any copyright or trademark symbols that usually accompany proprietary names, eg (R), (C), or TM  (e.g. next to drug or reagent names). Therefore please remove all instances of trademark/copyright symbols throughout the text, including "UMLS®" on page 12.

3. Please update the completed 'Competing Interests' statement, including any COIs declared by your co-authors. If you have no competing interests to declare, please state "The authors have declared that no competing interests exist". Otherwise please declare all competing interests beginning with the statement "I have read the journal's policy and the authors of this manuscript have the following competing interests:"

Additional Editor Comments (if provided):

Authors are requested to response to the reviewers comments - particularly related to the baseline.

Reviewers' comments:

Reviewer's Responses to Questions

**Comments to the Author**

1. Does this manuscript meet PLOS Digital Health’s publication criteria? Is the manuscript technically sound, and do the data support the conclusions? The manuscript must describe methodologically and ethically rigorous research with conclusions that are appropriately drawn based on the data presented.

Reviewer #1: Yes

Reviewer #2: Partly

2. Has the statistical analysis been performed appropriately and rigorously?

Reviewer #1: I don't know

Reviewer #2: N/A

3. Have the authors made all data underlying the findings in their manuscript fully available (please refer to the Data Availability Statement at the start of the manuscript PDF file)?

Reviewer #1: Yes

Reviewer #2: Yes

4. Is the manuscript presented in an intelligible fashion and written in standard English?

Reviewer #1: Yes

Reviewer #2: Yes

5. Review Comments to the Author

Reviewer #1: Innovation= Useful + Novel. The authors argue that the tool is very useful esp. for non English text. The question I have is whether the approach used is novel compared to what already exists. It would be good to clarify this in the narrative.

Also in the concluding section, it would be great to get an assessment of performance of the tool vs purely data-driven approaches in English text as well as other non English text.

Reviewer #2: Summary

+ This manuscript describes an open annotation framework, DrNote, which leverages public Wikipedia data to create a medical entity linker. This annotator builds on the OpenTapioca, an entity linking framework. OpenTapioca requires annotating documents to tag entity mentions. DrNote proposes a distantly supervised approach for creating the labeled training data (for symptoms, diagnoses, drugs and medications) by creating an annotated dataset using string matching with WikiData & Wikipedia. A key benefit of this approach is taking advantage of multiple languages in wikipedia, a key challenge area in clinical/EHR NLP. The resulting software and dataset are provided.

Strengths

+ Multilinguality is a critical application area and Wikidata has some considerable advantages for covering multiple languages

+ Building a WikiData KG subset of key medical concepts is a useful dataset contribution

+ Software is open source

Weaknesses

+ The primary weakness of this manuscript is that there are no empirical results which with to evaluate the performance of the proposed annotation framework. The manuscript focuses on describing the process by which the wiki dataset is created and various aspects of the general software platform (and how it uses/interfaces with OpenTapioca and Apache Solr) but there are no empirical evaluations or experiments to measure quality of the annotator. This is a significant weakness and makes it difficult to evaluate the merits of the proposed contributions.

+ The domain shift from wiki text to EHR/clinical text is likely quite significant. A spot check with some MIMIC-III data (see below) reveals some of the limitations, but this needs to be characterized systematically using expert-labeled datasets. Restricting to medical entities in the WikiData KB is a valid use case, (for example, for tagging medical concepts in web data or consumer facing health literature this might be fine) but the cost of the domain shift needs to be measured

+ The multilinguality capability, while appealing, isn't motivated by results on multilingual datasets.

+ There are commercial solutions that handle Multilinguality (e.g., Amazon Comprehend Medical). It would be nice to include a baseline compared to such a service.

Recommendations

+ The authors need to provide empirical measures of their systems performance by evaluating on some expert annotated (bio)medical datasets. Even an NER evaluation (vs. entity linking / NED, which is challenging here given the WikiData KG) would provide some sense of the annotator's term coverage and bound entity linking performance. There are a few parallel biomedical corpora that could be used:

- (biomedical) https://academic.oup.com/jamia/article/22/5/948/930067#210287674

- (biomedical) https://huggingface.co/datasets/scielo

For clinical text, the situation is quite sparse (as noted by the authors) but there are clinical corpora in English that could be used to assess NER/term coverage and at least provide some empirical measurements of transitioning from wiki data to clinical/EHR text. Something like the 2018 n2c2 Adverse Drug Event (ADE) and Medication Extraction Challenge would work fine as an evaluation here.

+ There is considerable prior work on distant/weakly supervised, pseudo/silver-labeling and other methods for automatically generating training data that should be discussed as background and used to highlight the strengths of this work. These approaches are especially common in clinical concept recognition, but (as the authors note) this area is under-explored in multilingual settings. There are a few nice surveys of recent methods

- "Modern Clinical Text Mining: A Guide and Review." Bethany Percha. Annual Review of Biomedical Data Science. 2021

- "Clinical Concept Extraction: a Methodology Review." Fu et al. Journal of Biomedical Informatics. 2020

- "Ontology-driven weak supervision for clinical entity classification in electronic health records." Fries et al. Nature Communications. 2021

Misc Comments

+ (Line 280) The 1-2 weeks doesn't provide a very meaningful estimate of compute costs since in conflates multiple sources of compute/time costs (e.g., download time, data preprocessing, indexing, SVM training). Does the pipeline benefit from multiprocessing or support a distributed/cluster setup? It would be more helpful to describe the compute costs more precisely and provide more details, e.g., some definition of throughput based on number of candidate entities processed, # of wiki pages, etc

Example Annotation Case

+ As a quick test, I ran this de-identified clinical text snipped from MIMIC-III on the public demo at https://textmining.misit-augsburg.de/

and using the filter rules for non-stopwords and min length of 2, we get the following results

"""

Reason: assess for gastric distention, gastric bubble

Admitting Diagnosis: GASTROINTESTINAL BLEED

66 [[year]] old [[man]] s/p R [[colectomy]] on [**3-9**] and [[CREST syndrome]] w/ ongoing [[nausea]].

A wedge-shaped area of increased density is seen on the lateral view, which I believe is corresponding to the [[left lower lobe]]. This has features of subsegmental [[atelectasis]], but a follow-up is recommended to evaluate for progression, as might be seen with a [[pneumonia]]. The remainder of the study shows no [[air]]-space [[disease]]. There is no evidence for congestive features nor pleural effusions.

"""

(enclosing double brackets indicate a detected entity , i.e., "[[ (entity) ]]")

We miss "gastric distention", "gastric bubble", "GASTROINTESTINAL BLEED", "pleural effusions", partially match "air-space disease" and "subsegmental atelectasis", and have a number of false positives ("year", "man")

6. PLOS authors have the option to publish the peer review history of their article (what does this mean?). If published, this will include your full peer review and any attached files.

**Do you want your identity to be public for this peer review?** For information about this choice, including consent withdrawal, please see our Privacy Policy.

Reviewer #1: No

Reviewer #2: No

---

## [Decision Letter · Decision Letter 1]

27 Apr 2022

PDIG-D-21-00039R1

DrNote: An open medical annotation service

PLOS Digital Health

Dear Dr. Frei,

Thank you for submitting your manuscript to PLOS Digital Health. After careful consideration, we feel that it has merit but does not fully meet PLOS Digital Health's publication criteria as it currently stands. Therefore, we invite you to submit a revised version of the manuscript that addresses the points raised during the review process.

We look forward to receiving your revised manuscript.

Kind regards,

Imon Banerjee

Section Editor

PLOS Digital Health

Journal Requirements:

Additional Editor Comments (if provided):

Congratulation to the authors for substantially improving the manuscript in the first revision. As per the suggestion of the reviewer 1, I would strongly suggest the authors to add comparative analysis with the existing/traditional baseline.

Reviewers' comments:

Reviewer's Responses to Questions

**Comments to the Author**

1. If the authors have adequately addressed your comments raised in a previous round of review and you feel that this manuscript is now acceptable for publication, you may indicate that here to bypass the “Comments to the Author” section, enter your conflict of interest statement in the “Confidential to Editor” section, and submit your "Accept" recommendation.

Reviewer #1: (No Response)

Reviewer #2: All comments have been addressed

2. Does this manuscript meet PLOS Digital Health’s publication criteria? Is the manuscript technically sound, and do the data support the conclusions? The manuscript must describe methodologically and ethically rigorous research with conclusions that are appropriately drawn based on the data presented.

Reviewer #1: Partly

Reviewer #2: Yes

3. Has the statistical analysis been performed appropriately and rigorously?

Reviewer #1: Yes

Reviewer #2: Yes

4. Have the authors made all data underlying the findings in their manuscript fully available (please refer to the Data Availability Statement at the start of the manuscript PDF file)?

Reviewer #1: Yes

Reviewer #2: Yes

5. Is the manuscript presented in an intelligible fashion and written in standard English?

Reviewer #1: Yes

Reviewer #2: Yes

6. Review Comments to the Author

Reviewer #1: The authors are yet to bring out uniqueness of this approach compared to other available applications. Only providing references is not sufficient.

Reviewer #2: The authors have nicely addressed all my concerns and have substantially improved the manuscript with the addition of the empirical evaluation.

7. PLOS authors have the option to publish the peer review history of their article (what does this mean?). If published, this will include your full peer review and any attached files.

**Do you want your identity to be public for this peer review?** For information about this choice, including consent withdrawal, please see our Privacy Policy.

Reviewer #1: No

Reviewer #2: No

---

## [Editor Report · Decision Letter 2]

12 Jul 2022

DrNote: An open medical annotation service

PDIG-D-21-00039R2

Dear Mr. Frei,

We are pleased to inform you that your manuscript 'DrNote: An open medical annotation service' has been provisionally accepted for publication in PLOS Digital Health.

Best regards,

Imon Banerjee

Section Editor

PLOS Digital Health